# The Role of Novel Agents in Treating CLL-Associated Autoimmune Hemolytic Anemia

**DOI:** 10.3390/jcm10102064

**Published:** 2021-05-12

**Authors:** Alessandro Noto, Ramona Cassin, Veronica Mattiello, Gianluigi Reda

**Affiliations:** 1Hematology Unit, IRCCS Ca’ Granda Ospedale Maggiore Policlinico, 20122 Milan, Italy; ramona.cassin@policlinico.mi.it (R.C.); gianluigi.reda@policlinico.mi.it (G.R.); 2Department of Oncology and Hemato-Oncology, University of Milan, 20122 Milan, Italy; veronica.mattiello@unimi.it

**Keywords:** CLL, autoimmune, hemolytic, anemia, ibrutinib, idelalisib, venetoclax

## Abstract

Autoimmune cytopenias (AICs) have been reported as a common complication in chronic lymphocytic leukemia (CLL) with autoimmune hemolytic anemia (AIHA), accounting for most cases. According to iwCLL guidelines, AICs poorly responsive to corticosteroids are considered indication for CLL-directed treatment. Chemo-immunotherapy has classically been employed, with variable results, and little data are available on novel agents, the current backbone of CLL therapy. The use of idelalisib in the setting of AICs is controversial and recent recommendations suggest avoiding idelalisib in this setting. Ibrutinib, through ITK-driven Th1 polarization of cell-mediated immune response, is known to produce an immunological rebalancing in CLL, which stands as a fascinating rationale for its use to treat autoimmunity. Although treatment-emergent AIHA has rarely been reported, ibrutinib has shown rapid and durable responses when used to treat AIHA arising in CLL. There is poor evidence regarding the role of BCL-2 inhibitors in CLL-associated AICs and the use of venetoclax in such cases is debated. Furthermore, their frequent use in combination with anti-CD20 agents might represent a confounding factor in evaluating their efficacy. In conclusions, because of their ability to mitigate an immunological dysregulation that is (at least partly) responsible for autoimmunity in CLL, to date BTK-inhibitors stand out as the most suitable choice when treatment of autoimmune cytopenias is required.

## 1. Introduction

Chronic lymphocytic leukemia (CLL), the most common leukemia in western countries, is characterized by a marked clinical heterogeneity, ranging from years of stable disease to a rapidly progressive clinical course [1]. Due to several concomitant immunologic dysregulations that are typical of this entity, patients are subject to an increased risk of secondary malignancies, infectious events and autoimmune complications [2]. Autoimmune cytopenias (AICs) have long been reported as a common complication in CLL with autoimmune hemolytic anemia (AIHA) accounting for most cases (7–10%) followed by immune thrombocytopenia (ITP) (1–5%), and less common disorders such as pure red cell aplasia (PRCA), and autoimmune granulocytopenia (AIG) [3].

According to 2018 iwCLL guidelines [4] autoimmune complications including anemia or thrombocytopenia poorly responsive to corticosteroids are considered indication for treatment.

The diagnosis of AIHA (in CLL patients) usually requires the presence of the following criteria [5,6]: hemoglobin (Hb) levels below 11 g/dL without other explainable etiology or recent cytotoxic therapy; evidence of hemolysis by one or more laboratory markers (high reticulocyte count, consumed serum haptoglobin levels, elevated lactate dehydrogenase (LDH), or unconjugated bilirubin levels); evidence of a positive direct antiglobulin test (DAT) for either IgG or C3 or the presence of cold agglutinins.

Similarly to primary cases, CLL-associated AIHA is generally categorized into two major entities: a warm type (wAIHA), characterized by a DAT positive to IgG or IgG + C3d and negative agglutination at 20 °C, and a cold type (cold hemagglutinin disease, CAD), with agglutination at 20 °C and a DAT positive for C3d. When available, a peripheral blood smear can sometimes be helpful as spherocytes are usually present in wAIHA, while red blood cell agglutination is a frequent feature of CAD.

Autoantibodies are usually polyclonal high-affinity IgGs produced by non-malignant self-reactive B-cells (90% of cases), while CLL cells may sometimes produce autoantibodies (mainly IgM) in <10% of cases [7].

Although it is usually possible to distinguish between the two forms, in some cases, a mixed DAT can be present, characterized by a DAT positive for IgG and C3, coexisting with high titre (>40) cold agglutinins [5].

Moreover, rare cases of DAT-negative AIHA in CLL patients have been described, possibly due to the low affinity or to the very small number of autoantibodies, and, on the other hand, DAT positivity is sometimes present in CLL patients without evidence of active hemolysis [8].

For cases arising in CLL, diagnosis can sometimes be challenging since some markers might be influenced by the underlying disease: LDH may be increased in progressive disease and haptoglobin might be only slightly reduced or normal when inflammatory/infective conditions are present; otherwise reticulocytosis might be impaired by bone marrow infiltration.

## 2. Standard Management of AIHA

It is important to underline that no prospective clinical trial has ever been conducted in the setting of CLL-associated AIHA, as patients with uncontrolled AICs are typically excluded from trials. Current strategies for first-line treatment are based on the approach used in primary AIHA.

When considering treatment, the acuteness of onset, the severity of anemia and the degree of hemolysis must be taken into account, together with patient’s symptoms, age and comorbidities. Blood transfusions are usually indicated with Hb < 6 g/dL or higher in elderly comorbid patients. Over-transfusion should be avoided since it carries a high risk of allo-immunization [3].

The management of CLL-associated wAIHA must take into account the stage of CLL: since isolated AIHA is not a criterion to treat CLL, patients in A stage should not receive CLL directed treatment. 

For warm AIHA occurring in patients with stage A CLL or in whom AIHA is the predominant feature, prednisone at 1 mg/kg/day for 3–4 weeks, followed by a slow tapering over several weeks (no less than 12 weeks) is considered the standard first line [9]. Methylprednisolone boli, with or without intravenous immunoglobulins, may be added in patients with acute hemolysis and a slow response to steroid therapy [10]. Rituximab can be considered the second line, with an overall response rate of 71% (10/14) in one study [11], although responses are often not sustained [2]. 

A retrospective study evaluating obinutuzumab in a small cohort of CLL patients showed an overall response rate (ORR) of 87.5% (7/8) (complete response (CR) 50%) demonstrating that 2nd generation anti-CD20 agents may be an effective treatment for AIHA and ITP in patients with CLL/SLL [12].

Some patients with CAD may have a milder clinical presentation with cold agglutinin associated symptoms (e.g., acrocyanosis, itching, urticaria) and not need immediate treatment. Therapy should be reserved for transfusion-dependent cases, active hemolysis, and significant CAD-related cAIHA symptoms. As corticosteroids result in remission in very few patients, rituximab should be considered as a first line. A standard dose of 375 mg/sm weekly for 4 weeks guarantees a response in up to 70–100% of patients [13,14].

Considering patients refractory to first-line treatment (both wAIHA and cAIHA, or those with active CLL) current guidelines advise the introduction of a CLL-directed therapy that should be tailored to disease features and the patient’s comorbidities [15]. 

Historically, in the setting of wAIHA related to B-lymphoproliferative malignancies, combination regimens such as rituximab, cyclophosphamide and dexamethasone (RCD) or rituximab, cyclophosphamide, vincristine and prednisolone (R-CVP) have shown satisfying ORR (>80%) with a response duration of approximately 22 months [16,17]. Subsequently, FCR (Fludarabine, Cyclophosphamide, Rituximab) and BR (Bendamustine Rituximab) were recognized as the standard of care as far as immunochemotherapy in CLL is concerned. Particularly, the combination of bendamustine and rituximab was found to be effective in warm AIHA, showing an overall response rate of 81% in 26 patients, and a time to next treatment of 28.3 months [18]. Fludarabine carries the risk of inducing AIHA or pure red cell aplasia and should be avoided [19], although this effect seems less pronounced when combined with rituximab.

Nowadays, with the availability of novel agents, treatment selection should be accurately made keeping in mind their ability to control disease burden and, possibly, to modulate autoimmunity in CLL.

## 3. Novel Agents and AIHA

With the increasing use of small molecule inhibitors in clinical practice, chemotherapy has a limited role in CLL treatment, but when it comes to addressing autoimmune complications, no extensive literature is available on their use for AIHA. 

While their anti-tumor effect is well known, not all novel agents have shown immunomodulating activities; such peculiar mechanisms may justify their use against autoimmune disorders.

There is currently no consensus on which agent should be preferred for AICs in CLL and the choice is largely left to single-center policy. 

Though literature is scarce, several centers have reported their experiences with small molecules currently in use, with interesting results (Table 1).

## 4. Idelalisib

Little data evaluating the efficacy of Phosphatidylinositol 3-kinase-δ (PI3K-δ) inhibitors in CLL-related AICs are available to date. In particular, idelalisib has demonstrated correlation to autoimmune clinical complications (i.e., pneumonitis, diarrhea and transaminitis) which does not make this class of inhibitors an attractive option for the treatment of AIHA. Therefore, recent recommendations suggest avoiding idelalisib in the context of CLL-associated cytopenias [29]. 

Limited clinical reports have shown some degree of efficacy of idelalisib in addressing this condition, although a confounding effect of rituximab should be considered.

A multicentre French study enrolled 44 CLL patients with AICs, of whom 25 received ibrutinib monotherapy and 19 patients received the idelalisib-rituximab association (12 patients were treated for AIHA).

Overall response rates in both groups were 95% and 92% respectively, regardless of AIC type. In the R-idelalisib group, the ORR was 92% for AIHA patients and 100% for patients with ITP or PRCA. 

Notably, therapy with kinase inhibitors (KIs) allowed discontinuation of corticosteroids in 86% of ibrutinib patients and in 67% in the R-idelalisib cohort. With a median follow-up of 26.8 months, treatment failed to control AIHA in one case in the idelalisib group. For responding patients, no AIHA relapse was described during idelalisib treatment, although five patients experienced relapse after idelalisib discontinuation [21].

Feld J et al. reported a case of unprecedented concomitance of mixed AIHA and B-prolymphocytic leukemia (PLL) evolution in an elderly CLL patient treated with a combination of prednisone, idelalisib and rituximab, allowing normalization of Hb levels and white blood count within 2 months of therapy [20].

Although these results seem to show CLL-related AIHA might be effectively treated by the use of idelalisib, we need to consider that the combination with rituximab and/or steroids makes it difficult to selectively evaluate its efficacy against autoimmune cytopenias.

## 5. Ibrutinib

There is growing evidence suggesting that Bruton’s Tyrosine Kinase (BTK) inhibitors may soon play an important role in the treatment of refractory AIHA in clinical practice. 

Ibrutinib is a first-in-class inhibitor of BTK approved for the treatment of treatment-naive (TN) and relapsed/refractory (R/R) CLL patients. As we will further elaborate, the many off-target effects of this agent provide a rationale for its use in the setting of CLL-related AICs [30,31]. However, an important caveat of most clinical trials using BTK-inhibitors is that they excluded patients with active AIC, which hampers the evaluation of this aspect on large groups. As a consequence, most observations consist of case reports and single-center experiences. 

A small case series of five patients with AIHA requiring therapy in previously treated CLL patients was reported by Garcia-Horton et al. All five patients showed a response to ibrutinib and AIHA was controlled within a median of 6.5 weeks, with discontinuation of steroids achieved in all patients at a median of 10 weeks; none of the five patients showed evidence of AIHA relapse at a median follow-up of 3 years [22].

Other similar case reports/series have been published over the last 5 years, confirming ibrutinib’s efficacy in treating AIHA in patients affected by CLL [23,25,32] and other B-cell lymphomas [33].

Perhaps the most extensive data so far available of CLL-related AIHA and ibrutinib are based on a sub-analysis of the phase III RESONATE trial (ibrutinib vs. ofatumumab in R/R CLL patients) evaluating the effects of this agent on patients with concomitant AIHA and/or ITP [26]. 

Although uncontrolled AIHA/ITP patients were excluded, the presence of mild autoimmune hemolysis in patients who met other iwCLL treatment criteria was allowed. 

Therefore, patients with ongoing AIHA (*n* = 21 ibrutinib; *n* = 9 ofatumumab) at study entry were evaluated. Early after treatment initiation median Hb levels started rising and were maintained throughout a median follow-up of 17.5 months. In this analysis no new onset AIHA occurred in the ibrutinib group compared to four patients on ofatumumab. This event has been only sporadically reported in few series of CLL patients [34].

Hampel et al., using the Mayo Clinical CLL Database, retrospectively analyzed 193 CLL patients treated with ibrutinib with focus on 12 patients who were treated for AIHA outside clinical trials. Eight (67%) patients were able to discontinue or de-escalate AIC treatment and no patients had worsening of their AIC after initiating ibrutinib; 11/193 patients (among whom were 5 cases of AIHA: 3 cases in patients with remote history of AIC and 2 cases in patients with no such history), however, experienced treatment-emergent AIC, leading to drug discontinuation in 36% of cases [24]. Notably, treatment-emergent autoimmune events were seen exclusively in patients with unmutated IGHV status, which notoriously correlates with a higher incidence of AICs [35].

To date, little evidence is available regarding novel BTK-inhibitors, such as acalabrutinib, and their effect on autoimmune complications in CLL, but results of a phase II study evaluating acalabrutinib monotherapy in 134 R/R CLL showed only one recurrence of AIHA among 11 patients with history of AIC [36].

All things considered, published data suggest BTK-inhibitors and particularly ibrutinib represent a safe and effective therapeutic strategy in patients with history of AIC.

## 6. Venetoclax

Venetoclax, a BCL-2 inhibitor, has demonstrated activity in R/R CLL patients regardless of TP53 disruption when used alone or associated with rituximab [37].

There is poor evidence regarding the role of BCL-2 inhibitors in treating autoimmune events in CLL as there are no trials addressing this issue. Nonetheless, some cases of emerging AHIA/ITPs have occasionally been reported in clinical trials.

In particular, in their trial evaluating single-agent venetoclax in untreated 17p-deleted patients, Stilgenbauer et al. observed 8/158 patients experiencing AHIA and 3/158 immune thrombocytopenia during treatment, although none needed dose interruptions [38]. 

Notably, all patients enrolled in the study harbored high-risk cytogenetic aberrations that are known to be associated with a higher frequency and severity of autoimmune cytopenias [39].

This event, however, was not observed in other trials where venetoclax was used in combination with anti-CD20 agents, possibly due to an additional immunosuppressive effect [40,41]. 

Though limited to single case reports, some authors have reported their experience of successfully treating AICs with venetoclax. 

In 2017 Lacerda et al. described the case of a R/R 17p-deleted patient treated with alemtuzumab for concomitant disease progression and AHIA, with disease control but persistence of transfusion-dependent anemia. After failure with rituximab and high-dose dexametasone he was started on compassionate-use venetoclax with a significant improvement in Hb levels after 3 months and no relapse of AHIA after 10 months of therapy [27].

Gordon et al. reported 2 cases (one ITP and one Evans Syndrome) that received single-agent venetoclax after several relapses of isolated AICs with little disease burden: both patients had a normalized Hb and platelet count within 4 weeks of starting therapy [28].

BCL-2 inhibitors might represent a potential option in patients with refractory CLL-associated AHIA. In the absence of data comparing different small molecules inhibitors, it is debated whether venetoclax may be considered a preferred choice in this setting. 

## 7. Discussion

AIHA is the most common AIC in CLL patients, with a prevalence of approximately 2.9% in stable Binet stage A disease compared with 10.5% in stage B and C [2]. 

When AIHA occurs in CLL patients, acquiring information on the type of AIHA (warm, cold or mixed), biological CLL features based on FISH and IGHV, and the presence of other iwCLL treatment criteria is considered an essential step.

Considering the suboptimal response to chemo-immunotherapy (CIT) regimens in high risk CLL patients (IGHV unmutated, del 17p/TP53) [42] together with the incidence of AIC in CLL, it is of interest to assess the role of small molecule inhibitors (i.e., idelalisib, venetoclax, and ibrutinib) in the context of CLL-related AIHA. 

Nowadays, data regarding the use of targeted agents derive from retrospective studies, and direct comparisons between different small molecule inhibitors in this group of patients are lacking. As a consequence, no guidelines are available to direct the choice of therapy in CLL-patients developing AIHA and not responding to steroids/rituximab and chemo-immunotherapy. 

Currently available data show that ibrutinib appears to cause an improvement in the majority of patients with pre-existing AIHA [22,23,25,26,33], probably as a result of an immunological re-balancing that is not mediated by BTK blockade. 

In addition to its direct anti-tumor effect on BTK, ibrutinib irreversibly binds ITK (Interleukin-2 inducible kinase) and inhibits downstream activation of Th2 cells after TCR stimulation both in vitro and in vivo [31]. As ITK disruption can contribute to the pathogenesis of certain infectious, autoimmune, and neoplastic diseases, this “side” activity has represented an interesting point of speculation. It has been demonstrated that CD4 T-cell populations isolated from CLL patients are skewed at a molecular and phenotypic level towards a Th1 profile after a brief in vitro dose of ibrutinib, confirming a T-cell-specific effect of this agent [30,31]. These results are in line with the assumption that ibrutinib is a potent and clinically relevant immuno-modulating drug, thus showing the potential to repurpose this therapy for other diseases that result from or cause a disproportionate polarization of Th2 immunity.

Although concerns have been raised because of reports of new onset AIHA after ibrutinib start [34], current evidence seems to show that the potential benefits in terms of efficacy may compensate the low risk of hemolysis recurrence.

Moreover, in most patients where ibrutinib is used for the treatment of AIHA, time to steroid interruption is generally shorter than usual practice (10–12 weeks). As tapering off corticosteroids can often requires up to 6 months, patients are prone to experience several well-known complications of this treatment (opportunistic infections, metabolic syndrome, osteoporosis, and psychiatric symptoms). Therefore, especially in patients with coexisting conditions, shortening steroid exposure represents an attractive option in patients who would otherwise be subject to further therapy-induced complications.

For these reasons, we tend to favor ibrutinib in our practice for the treatment of CLL-associated AIHA.

Idelalisib is frequently responsible for triggering non-hematological autoimmune complications, whereas data on its use in the management of AICs are limited. 

Although idelalisib in association to rituximab has shown some degree of efficacy in the treatment of CLL-related AIHA [20,21], the incidence of potentially severe side effects and shorter progression free survival [43], do not make it a first-choice option among small molecules. 

Furthermore, the beneficial effects of co-administering an anti-CD20 monoclonal antibody with a targeted drug to gain a better control of pre-existing AIHA still need to be clarified.

Finally, with regard to the potential use of BCL-2 inhibitors, the role of venetoclax in the treatment of AIHA in CLL is yet to be defined.

## 8. Conclusions

Small molecule inhibitors are able to exert excellent disease control in chronic lymphocytic leukemia as first-line options or in R/R cases. Such activity on disease burden is usually able to extinguish the autoimmune dysfunction that is responsible for AIHA and related immune cytopenias. However, the choice of the right agent seems important as, with very little literature available so far, not all new drugs are able to yield the same efficacy in repressing hemolysis. Because of their ability to produce a deep change in the cytokine asset that is (at least partly) responsible for autoimmunity in CLL, to date BTK-inhibitors stand out as the most suitable choice.

In the near future, clinical studies employing BTK-inhibitors as treatments for CLL-related AICs will manage to cast new light on their potential role as an earlier therapy line for these conditions and phase II trials using these inhibitors (with or without rituximab) are currently ongoing in CLL-wAIHA (NCT03827603) and ITP (NCT03395210).

## Figures and Tables

**Table 1 jcm-10-02064-t001:** Autoimmune hemolytic anaemia; characteristics, management, and outcome in CLL patients treated with novel agents.

Treatment	Line for AIHA	*n* of Patients	ORR (%)	DOR (Months)	Reference
R-Idelalisib/Ibrutinib *	1–6	12/16	92/87.5	26.8	Godet S et al. 2018 [20]
R-Idelalisib + PDN	1	1	NA	3	Feld J et al. 2019 [21]
Ibrutinib	2	5	100	32.5	Garcia-Horton A et al. 2018 [22]
Ibrutinib	5	1	NA	12	Cavazzini et al. 2016 [23]
Ibrutinib **	>1	29	67	29	Hampel PJ et al. 2018 [24]
Ibrutinib +/− Rituximab *	1–4	8	75	17.6	Vitale C et al. 2016 [25]
Ibrutinib +/− steroids *	1–2	21	NA	17.5	Montillo M et al. 2017 [26]
Venetoclax	4	1	NA	10	Lacerda MP et al. 2017 [27]
Venetoclax *	3	2	NA	6	Gordon MJ et al. 2019 [28]

* Studies evaluating both AIHA and ITP, all data presented are focused only on AIHA patients. ** Data regarding all AICs included in the study. NA: not available; DOR: Duration of Response; AIHA: autoimmune hemolytic anemia.

## Data Availability

No new data were created or analyzed in this study.

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
