# Peer review of "The Role of Novel Agents in Treating CLL-Associated Autoimmune Hemolytic Anemia"

_jcm, 2021, doi:10.3390/jcm10102064_

Round 1

Reviewer 1 Report

The manuscript provides an interesting and comprehensive review of the role of small molecule inhibitors in the treatment of auto-immune complications that can develop in patients with CLL.

There were 2 minor points to amend.

Line 111 (?word missing). small molecule 'inhibitor' 

Line 202. Author's name should be 'Stilgenbauer'

Author Response

Line 111 (?word missing). small molecule 'inhibitor'.  Revised as requested.

Line 202. Author's name should be 'Stilgenbauer' Revised as requested

Reviewer 2 Report

Initial therapy of isolated autoimmune hemolytic anemia (AIHA) associated with chronic lymphocytic leukemia (CLL) is similar to therapy of primary AIHA with steroids in warm and rituximab in the cold subtypes.   Therapy of relapsed CLL-associated hemolytic anemia involves CLL directed treatment and historically included combination chemoimmunotherapy.   As targeted therapy is becoming a mainstay of therapy of CLL, it is important to be familiar with the role of these agents in treatment of CLL associated autoimmune hemolytic anemia.  This manuscript provides a contemporary review of the data on use of the novel targeted agents in treatment of CLL-associated AIHA.

After a brief review of the initial therapy of CLL-associated AIHA and efficacy of anti-CD20 agents and combination chemoimmunotherapy the authors of this paper summarize currently available literature on efficacy and safety of novel agents in CLL-associated AIHA and provide rationale for use of BTK inhibitors based on their immunomodulatory properties in addition to CLL disease control.

The topic is pertinent to clinical hematology and the review is informative and very instructive.  It provides an important review of treatment autoimmune hemolytic anemia complicating CLL in the era of the novel therapies.  The article is well organized. The table is well organized and informative, providing a concise summary of the main studies reviewed.  

There are a few minor points that would provide more clarity to this important paper.

Line 58 “DAT positivity does not necessarily correlate with AIHA… Please clarify – development of AIHA? Presence of AIHA?

Line 111 “With the increasing use of small molecule in clinical practice… no extensive literature is available on their use for AIHA” Typographic error.  Consider rewriting this sentence for clarity.

Line 114 “While their anti-tumor effect is well known, not all of them have shown immunomodulating activities; such peculiar mechanisms may justify a rationale for their use...” specify “their.”

Line 136 Please define KI (has not been defined before)

Line 130-140 Please provide reference.  Should be reference 22.  Reference 21 provided corresponds to case report on prolymphocytic leukemia described by Feld J et al.

Line 186 Please comment how many cases developed autoimmune hemolytic anemia.

Line 186-187 “Of note, treatment‐emergent autoimmune events (such 187 cases?) were…” Typographic error.

Line 235 Typographic error

Author Response

Line 58 “DAT positivity does not necessarily correlate with AIHA… Please clarify – development of AIHA? Presence of AIHA? –> Thank you for pointing that out. It was meant as development of AIHA. We will rewrite this last part as follows “DAT positivity is sometimes present in CLL patients without evidence of active hemolysis”.

Line 111 “With the increasing use of small molecule in clinical practice… no extensive literature is available on their use for AIHA” Typographic error.  Consider rewriting this sentence for clarity. –> Thank you. We added “small molecule inhibitors”.

Line 114 “While their anti-tumor effect is well known, not all of them have shown immunomodulating activities; such peculiar mechanisms may justify a rationale for their use...” specify “their.” We will rephrase “While their anti-tumor effect is well known, not all novel agents have shown immuno-modulating activities”.

Line 136 Please define KI (has not been defined before). Thanks, revised as requested.

Line 130-140 Please provide reference. Should be reference 22.  Reference 21 provided corresponds to case report on prolymphocytic leukemia described by Feld J et al. I see reference 21 and 22 have been swapped, We will swap them back.

Line 186 Please comment how many cases developed autoimmune hemolytic anemia. Thank you for pointing that out. We will provide additional details rephrasing as follows “11/193 patients (among whom 3 cases of AIHA in patients with remote history of AIC and 2 cases of AIHA in patients with no such history) ...”

Line 186-187 “Of note, treatment‐emergent autoimmune events (such 187 cases?) were…” Typographic error. We will erase the part between brackets.

Line 235 Typographic error We will rephrase as follows data regarding the use of targeted agents ...

Reviewer 3 Report

Thank you for allowing me to review the manuscript by Noto et al., entitled The Role of Novel Agents in Treating CLL-Associated Autoimmune Hemolytic Anemia.  I believe that this is a timely and relevant article, though understand that there is limited data (i.e. no randomized trials), to truly inform treatment decisions.  Overall, I believe the authors did a valiant job presenting the evidence and synthesizing conclusions.  Please see below for suggested revisions.  Rather than divide my recommendations into large/minor, I will go chronologically instead.

Page 1.

Line 13-14, sentence starting with “Chemo-immunotherapy” seems to be missing the word “the” prior to the word “current.”

Line17: “Although treatment-emergent AIHA…”  Ibrutinib has shown rapid and durable responses?  Or ibrutinib has led to rapid and durable remission of CLL-associated AIHA?  This sentence is not clear.

Line 20: “No trials addressing this issue,” meaning no trials addressing issue of BCL-2 inhibitors in CLL-associated AICs.  This implies that there are trials looking at PI3K inhibitors and BTK inhibitors in CLL-associated AICs, which is not true.

Page 2.

Line 46: “Similar...: a warm type (wAIHA), characterized…(fix wording); change similarly to similar and revealed to characterized.

Page 2 paragraph 1: This is OK but very general and of course multiple exceptions

Should you mention a smear?  In WAHA usually see spherocytes, in CAD you see agglutination.  This can be very helpful in distinguishing the two.

Line 58: “DAT positivity does not necessarily correlate with AIHA.”  Do you mean that the strength of the DAT does not necessarily correlate to degree of hemolysis?  If this is what you mean, please be clear.

Line 60: Even if haptoglobin is increased as an acute phase reactant, it usually is decreased with true hemolysis.

Line 66: Typically instead of systematically

Line 70: I would be careful about giving thresholds for blood transfusions as this is very dependent on the clinical picture (though I know you say “usually”).

Line 77: Define what you mean by “full-dose prednisone.”  If 1mg/kg daily then say that

Line 77: We hardly *ever* taper prednisone over a full 6 months.  Usually much quicker than that

Line 86: Where do you get the information that the “fewer patients with CAD may have a milder clinical presentation?”  In my experience, CAD can be quite bad and difficult to control.  Moreover, even in high doses steroids are usually not effective in CAD unless you are actually using them to treat the underlying CLL.

Line 88: “Invalidating” is the wrong word choice

Page 3:

Line 100: I’m not sure I would use “recently” to describe FCR

Line 114: “While their anti-tumor effect is well known, not “all small molecules,” have shown…”

Line 136: Define “KI”

Line 137: Sentence starting “With a median follow-up…” is a run-on sentence and very difficult to follow.

Page 4.

Line 154: Please define TN/RR if this is your first time using the acronym in this paper.

Line 178:  I found the 4 paragraphs starting with “Interestingly, although” incredibly difficult to follow.  In the paragraphs before you talk about how ibrutinib may be helpful for AICs.  But then you switch to talking about how ibrutinib may induce AICs.  Moreover, are you talking about all AICs or just AIHA?  Your paper is focusing on AIHA, so I would make sure to highlight this particular type of AIC.  All I all, it feels choppy and difficult to follow.   After reading the section on ibrutinib, I was not convinced that this drug is helpful in CLL-associated AIHA; if that’s your point, make sure you support it with data and that it comes across more clearly.  Finally, I would re-order so you talk about PI3K inhibitors first, then venetoclax, then ibrutinib last.  This is how you build an argument—debunk the other two and then support the “best” choice.

Line 196: This first sentence about venetroclax is unclear.  It sounds like you are saying the TP53 disruption is associated with rituximab, but of course you are talking about use of venetoclax in TP53-mutated CLL

Page 5.

Line 208: Be careful about making such a bold statement, sentence starting with “This event.”

Line 221: Don’t talk about “ramp up” without describing what this means

Line 235: The paragraph starting with “Nowadays, data regarding” should be re-written as it is unclear.  Try to be concise!

Page 6.

Line 254: “Although concerns have been issues…” This sentence needs to be reworded.  Would not say “abundantly”—none of this data is “abundant.” Risk of recurrence of CLL or AIHA?  Again this is very unclear.

Other points:

  1. I would recommend discussing ongoing clinical trials looking at BTK inhibitors in primary ITP.  There may be trials looking at BTK inhibitors in primary AIHA moving forward.  This would support your thesis. NCT03395210
  2. We know that complement is important in CAD and probably to some degree in wAIHA (depending on what type of IgG is deposited on the cell surface and if it fixed complement). Is there any data in regard to BTK inhibitors and complement?  I would mention this.

Author Response

Line 13-14, sentence starting with “Chemo-immunotherapy” seems to be missing the word “the” prior to the word “current.” Thank you, edited as suggested.

Line17: “Although treatment-emergent AIHA…” Ibrutinib has shown rapid and durable responses? Or ibrutinib has led to rapid and durable remission of CLL-associated AIHA? This sentence is not clear. We will rephrase as follows “… Although treatment-emergent AIHA have rarely been reported, ibrutinib has shown rapid and durable responses when used to treat AIHA arising in CLL”

Line 20: “No trials addressing this issue,” meaning no trials addressing issue of BCL-2 inhibitors in CLL-associated AICs. This implies that there are trials looking at PI3K inhibitors and BTK inhibitors in CLL-associated AICs, which is not true. I see your point; I think it might be better to cut this last statement simply stating that “There is poor evidence regarding the role of BCL-2 inhibitors in CLL-associated AICs … and the use of venetoclax in such cases is debated”.

Page 2.

Line 46: “Similar...: a warm type (wAIHA), characterized…(fix wording); change similarly to similar and revealed to characterized. Thank you, edited as suggested

Page 2 paragraph 1: This is OK but very general and of course multiple exceptions.

Should you mention a smear? In WAHA usually see spherocytes, in CAD you see agglutination. This can be very helpful in distinguishing the two. We added the sentence “When available, a peripheral blood smear can sometimes be helpful as spherocytes are usually present in wAIHA, while red blood cell agglutination is a frequent feature of CAD”.

Line 58: “DAT positivity does not necessarily correlate with AIHA.” Do you mean that the strength of the DAT does not necessarily correlate to degree of hemolysis? If this is what you mean, please be clear. Thank you for pointing out this ambivalence; it was meant as development of AIHA. We would rewrite this last part stating “DAT positivity is sometimes present in CLL patients without evidence of active hemolysis”.

Line 60: Even if haptoglobin is increased as an acute phase reactant, it usually is decreased with true hemolysis. I see your point although we sometimes encounter patients with normal or slightly decreased haptoglobin when CLL is progressive or other inflammatory conditions are present I would re-phrase “… some markers might be influenced by the underlying disease: LDH might be increased in progressive disease and haproglobin might be only slightly reduced or normal when inflammatory/infective conditions are present”

Line 66: Typically instead of systematically Thank you, edited as suggested.

Line 70: I would be careful about giving thresholds for blood transfusions as this is very dependent on the clinical picture (though I know you say “usually”). We agree thresholds can be difficult to establish since many patient-related factors must be taken into account (e.g. presence of ischemic cardiac disease). We used usually” to refer to what is done in clinical practice, as also assessed by Fattizzo et al. in “Autoimmune Cytopenias in Chronic Lymphocytic Leukemia: Focus on Molecular Aspects”.

Line 77: Define what you mean by “full-dose prednisone.” If 1mg/kg daily then say that. We will add “prednisone at at 1 mg/kg day”

Line 77: We hardly *ever* taper prednisone over a full 6 months. Usually much quicker than that.

Thank you for analyzing this point. According to Jager at al. , altough in one study of 33 primary AIHA cases, relapse was more common if PDN was stopped in less than 6 months, the authors suggest that clinicians should begin to taper after 21 days from prednisone start and consider stopping treatment after at least 3 months. We will rephrase “ … followed by a slow tapering over several weeks (no less than 12 weeks).”

Line 86: Where do you get the information that the “fewer patients with CAD may have a milder clinical presentation?” In my experience, CAD can be quite bad and difficult to control. Moreover, even in high doses steroids are usually not effective in CAD unless you are actually using them to treat the underlying CLL. We were probably not very clear; we meant patients with CAD may not always need immediate treatment and when anemia is mild, it might be reasonable to only treat those with symptomatic anemia or disabling Raynaud-like symptoms. So, it will be rephrases “Some patients with CAD may have a milder clinical presentation with cold agglutinin associated symptoms (e.g. acrocianosys, itching, urticaria) and not need immediate treatment. Therapy should be …” I will erase the statement on high-dose corticosteroids rephrasing “As corticosteroids result in remission in very few patients, rituximab should be considered as a first line. ”

Line 88: “Invalidating” is the wrong word choice I think “significant” may be better suited.

Page 3:

Line 100: I’m not sure I would use “recently” to describe FCR. Thank you, we will rephrase with “subsequently, FCR and BR were recognised as standard of care in CLL …”

Line 114: “While their anti-tumor effect is well known, not “all small molecules,” have shown…”. Thank you, edited as suggested.

Line 136: Define “KI” . Edited as suggested

Line 137: Sentence starting “With a median follow-up…” is a run-on sentence and very difficult to follow.

We will add a period rephrasing it as follows; “With a median follow-up of 26.8 months, treatment failed to control AIHA in one case in the idelalisib group. For responding 138 patients, no AIHA relapse was described during treatment, although five patients experienced relapse after idelalisib discontinuation”

Page 4.

Line 154: Please define TN/RR if this is your first time using the acronym in this paper. Edited as suggested.

Line 178: I found the 4 paragraphs starting with “Interestingly, although” incredibly difficult to follow. In the paragraphs before you talk about how ibrutinib may be helpful for AICs. But then you switch to talking about how ibrutinib may induce AICs. Moreover, are you talking about all AICs or just AIHA? Your paper is focusing on AIHA, so I would make sure to highlight this particular type of AIC. All I all, it feels choppy and difficult to follow. After reading the section on ibrutinib, I was not convinced that this drug is helpful in CLL-associated AIHA; if that’s your point, make sure you support it with data and that it comes across more clearly. Finally, I would re-order so you talk about PI3K inhibitors first, then venetoclax, then ibrutinib last. This is how you build an argument—debunk the other two and then support the “best” choice.

Thank you for pointing that out. We will rephrase as follows: “Therefore, patients with ongoing AIHA (n = 21 ibrutinib; n = 9 ofatumumab) at study entry, were evaluated. Early after treatment initiation median Hb and platelet levels started rising and were maintained throughout a median follow-up of 17.5 months [...] In this analysis no new onset AIHA occurred in the ibrutinib group compared to four patients on ofatumumab. This event has been only sporadically reported in few series of CLL patients [31]. ”

Also, we will re-order the paragraphs discussing ibrutinib last (in the POC so as not to have most revisions misaligned).

Line 196: This first sentence about venetroclax is unclear. It sounds like you are saying the TP53 disruption is associated with rituximab, but of course you are talking about use of venetoclax in TP53-mutated CLL

We will rephrase “Venetoclax, a BCL2-inhibitor, has demonstrated activity in R/R CLL patients regardless of TP53 dysruption WHEN USED alone or associated with rituximab”

Page 5.

Line 208: Be careful about making such a bold statement, sentence starting with “This event.”

We rephrased as follows “ … anti-CD20 agents, possibly due to an additional immunosuppressive effect”

Line 221: Don’t talk about “ramp up” without describing what this means.

We will erase the part between brackets.

Line 235: The paragraph starting with “Nowadays, data regarding” should be re-written as it is unclear. Try to be concise!

Thank you, we rephrased as follows “Nowadays, data regarding the use of targeted AGENTS derive from retrospective studies and direct comparisons between different small molecules INHIBITORS in this group of patients are lacking. As a consequence, no guidelines are available to direct the choice of therapy in CLL patients developing AIHA and not responding to steroids/rituximab and chemo-immunotherapy”.

Page 6.

Line 254: “Although concerns have been issues…” This sentence needs to be reworded. Would not say “abundantly”—none of this data is “abundant.” Risk of recurrence of CLL or AIHA? Again this is very unclear.

We will rephrase with “Although concerns have been RAISED because of reports of new onset AIHA after ibrutinib start [31], current evidence seems to show that potential benefits in terms of efficacy may compensate the low risk of recurrence of hemolysis.”

Other points:

1.I would recommend discussing ongoing clinical trials looking at BTK inhibitors in primary ITP. There may be trials looking at BTK inhibitors in primary AIHA moving forward. This would support your thesis. NCT03395210. Thank you for the suggestion. We rephrased as followsIn the near future, clinical studies employing BTK-inhibitors as treatments for CLL- related AICs will manage to cast new light on their potential role as an earlier therapy line for these conditions and phase II trials using these inhibitors (with or without rituximab) are currently ongoing in CLL-wAIHA [NCT03827603] and ITP [NCT03395210]”.

Complement is important in CAD and probably to some degree in wAIHA (depending on what type of IgG is deposited on the cell surface and if it fixed complement). Is there any data in regard to BTK inhibitors and complement? I would mention this. To our knowledge, there are no studies addressing the potential role of ibrutinib modulating complement cascade.

Round 2

Reviewer 3 Report

Acceptable for publication.

(Acrocyanosis is spelled incorrectly in the text).